# Uplift and denudation history of the Ellsworth Mountains: insights from low temperature thermochronology

Joaquín Bastías-Silva[1,2,3], David Chew[1], Fernando Poblete[4], Paula Castillo[5], William Guenthner[6], Anne Grunow[7], Ian W.D. Dalziel[8], Airton N. C. Dias[9], Cristóbal Ramírez de Arellano[10], Rodrigo Fernandez[7]

[1] Department of Geology, Trinity College Dublin, College Green, Dublin 2, Ireland.
[2] Escuela de Geología, Facultad de Ingeniería, Universidad Santo Tomás, Santiago, Chile.
[3] Institute of Geochemistry and Petrology, Department of Earth Sciences, ETH Zürich, Switzerland.
[4] Departamento de Geología, Universidad de Chile, Santiago.
[5] Institut für Geologie und Paläontologie, Westfälische Wilhelms-Universität, Münster, Germany.
[6] Department of Geology, University of Illinois Urbana-Champaign, Urbana, IL, USA.
[7] Byrd Polar and Climate Research Center, Ohio State University, Columbus, USA.
[8] Institute for Geophysics, Jackson School of Geosciences, The University of Texas, USA.
[9] Departamento de Física, Química e Matemática, CCTS, UFSCar - Campus Sorocaba, Brazil.
[10] Carrera Geología, Facultad de Ingeniería, Universidad Andres Bello, Chile.

*Correspondence to*: Joaquín Bastías-Silva (j.bastias.silva@gmail.com)

**Abstract.** While thermochronological studies have constrained the landscape evolution of several of the crustal blocks of West and East Antarctica, the tectono-thermal evolution of the Ellsworth Mountains remains relatively poorly constrained. These mountains are among the crustal blocks that comprise West Antarctica and exhibit an exceptionally well-preserved Palaeozoic sedimentary sequence. Despite the seminal contribution of Fitzgerald and Stump (1991), who suggested an Early Cretaceous uplift event for the Ellsworth Mountains, further thermochronological studies are required to improve the current understanding of the landscape evolution of this mountain chain. We present new zircon (U-Th)/He (ZHe) ages, which provide insights into the landscape evolution of the Ellsworth Mountains. The ZHe ages collected from near the base and the top of the sequence suggest that these rocks underwent burial reheating after deposition. A cooling event is recorded during the Jurassic–Early Cretaceous, which we interpret as representing exhumation in response to rock uplift of the Ellsworth Mountains. Moreover, our results show that, while ZHe ages at the base of the sequence are fully reset, towards the top ZHe are partially reset. Uplift and exhumation of the Ellsworth Mountains during the Jurassic–Early Cretaceous was contemporaneous with the rotation and translation of this crustal block with respect to East Antarctica and possibly the Antarctic Peninsula. Furthermore, this period is characterised by widespread extension associated with the disassembly and breakup of Gondwana, with the Ellsworth Mountains playing a key role in the opening of the far South Atlantic. Based on these results, we suggest that uplift of the Ellsworth Mountains during the disassembly of Gondwana provides additional evidence for major rearrangement of the crustal blocks between the South American, African, Australian and Antarctic plates. Finally, uplift of the Ellsworth Mountains commenced during the Jurassic, which predates the Early Cretaceous uplift of the Transantarctic Mountains. We suggest that the rift-related exhumation of the Ellsworth Mountains occurred throughout two events: (i) a Jurassic uplift associated with

the disassembly of southwestern Gondwana and (ii) an Early Cretaceous uplift related with the separation between Antarctica and Australia, which is also recorded in the Transantarctic Mountains.

## 1 Introduction

The Ellsworth Mountains extend for ~350 km between the Transantarctic Mountains and the Antarctic Peninsula and are ~50 km wide (Fig. 1). They are located within West Antarctica, which is composed of crustal blocks that amalgamated along the
Pacific margin of Gondwana during the latest Precambrian to middle Phanerozoic (e.g. Dalziel and Elliot, 1982; Dalziel and Lawver 2001; Jordan et al., 2020; Riley et al., 2023). Furthermore, the Ellsworth Mountains form part of the most isolated and enigmatic crustal block (the Ellsworth-Whitmore Mountains block) in West Antarctica (Schopf, 1969; Dalziel and Elliot, 1982). Their exposures are comprised of a series of nunataks and mountains, which are dominated by the Heritage and Sentinel ranges (Fig. 1). Although the Antarctic icecap is intensively developed at these latitudes, the Ellsworth Mountains yield an
extensive sedimentary record that extends from the Neoproterozoic to the Permian (Fig. 2; Craddock, 1969; Webers et al., 1992a; Castillo et al., 2017). Several studies have assessed the geological evolution of the Ellsworth Mountains along with its paleogeographic significance (Curtis et al., 1999; Curtis, 2001; Flowerdew et al., 2007; Dalziel, 2007; 2014, Castillo et al., 2017, Craddock et al., 2017). However, its tectonothermal evolution is relatively poorly constrained with the exception of the study of Fitzgerald and Stump (1991), who reported an Early Cretaceous uplift event based on apatite fission-track analyses.
Most studies concerning the tectonothermal history of West Antarctica have instead been conducted on the Antarctic Peninsula (e.g. Guenthner et al., 2010; Twinn et al., 2022; Bastias et al., 2022) and Thurston Island (e.g. Zundel et al., 2019). Further efforts to constrain the regional landscape evolution have been undertaken in the Transantarctic Mountains (e.g. Fitzgerald, 1994; Fitzgerald and Stump, 1997), which extends for ~3,000 km and divide East from West Antarctica (e.g. Goodge, 2020). To better understand the thermal evolution of the Ellsworth Mountains, we present herein new zircon (U-Th)/He data to
constrain its thermal history and hence the formation of its present-day landscape. Furthermore, the (U-Th)/He is a thermochronometric system that is sensitive to low temperatures (Wolf et al., 1996) and has the potential to provide robust constraints on the thermal evolution of basins along with their subsequent exhumation histories (e.g. Ault et al., 2019; Dai et al., 2019).

## 2 Ellsworth Mountains

The Ellsworth Mountains hosts a stratigraphic sequence that spans the Palaeozoic era and is up to ~13 km thick (Fig. 2; e.g. Webers et al., 1992a; Castillo et al., 2017). At the base of the sequence is the lower Palaeozoic Heritage Group (Webers et al., 1992b), which consist of ~7.5 km of strata that are almost exclusively present in the Heritage Range (Fig. 2). They consist of sedimentary and volcanic rocks that were deposited in a rapidly subsiding basin (e.g. Curtis and Lomas, 1999). The group is composed, from base to top, by the Union Glacier, Hyde Glacier, Drake Icefall, Conglomerate Ridge, Liberty Hills, Springer

Peak, Frazier Ridge and Minaret formations (Fig. 2). The Union Glacier Formation includes continental volcanic and volcaniclastic rocks (~3 km thick) and its age is constrained by a U-Pb zircon age from a Cambrian hyaloclastite (512 ± 14 Ma; personal communication, Rees et al., 1998). However, two metavolcaniclastic rocks from this formation yield U-Pb zircon ages of ~675 Ma, raising questions as to the depositional age of this unit (Castillo et al 2017). The Hyde Glacier Formation locally overlies the Union Glacier Formation and is composed of fluvial to shallow-marine deltaic deposits (Webers et al.,

1992b). Overlying these formations is the Drake Icefall Formation, comprised of black shales interbedded with limestones deposited in a shallow-marine environment (Jago and Webers, 1992). Conglomeratic quartzite and polymictic conglomerates of the Conglomerate Ridge Formation (Webers et al., 1992b) structurally overlie the Drake Icefall Formation with their contact defined by a reverse fault. Three laterally equivalent formations overlie the Conglomerate Ridge Formation. Termed the Springer Peak, Liberty Hills and Frazier Ridge formations. They are mostly clastic in composition and are comprised of

argillite, graywacke and quartzite (Webers et al., 1992b). Locally overlying these deposits is the Minaret Formation (Curtis and Lomas, 1999), which is dominated by marble and carbonate rocks deposited during the Late Cambrian (e.g. Buggisch and Webers, 1992; Jago and Webers, 1992). The Transitions Beds represent the uppermost unit of the Heritage Group and are comprised of by a thin succession of sandstone interbedded with argillite (Spörli, 1992).

Overlying the Heritage Group is the Crashsite Group, a ~3 km-thick sequence dominated by quartzite, argillite conglomerate,

limestone and basic volcanic rocks (Fig. 2; Goldstrand et al., 1994; Spörli, 1992). This sequence is comprised from bottom to top by the Howard Nunataks, Mount Liptak and Mount Wyatt Earp formations, which were deposited in a shallow-marine to fluviatile environment (Curtis and Lomas, 1999; Spörli, 1992). The age of the Crashsite Group has been constrained to the late Cambrian–Devonian by trilobite faunas, sedimentation rates and detrital zircon ages (Shergold and Webers, 1992; Spörli, 1992; Webers et al., 1992b; Flowerdew et al., 2007).

The Crashsite Group is conformably overlain by the Whiteout Conglomerate (Fig. 2). These rocks are 1 km thick and dominated by late Carboniferous to early Permian grey to black diamictites, which are associated with the Permo–Carboniferous Gondwanan glaciation (Matsch and Ojakangas, 1992). Overlying the Whiteout Conglomerate is a 1 km thick sequence of argillites, siltstone, sandstone and coal of the Polarstar Formation (Fig. 2; Collinson et al., 1992). Detrital zircons geochronology implies this unit was deposited during the Permian (Elliot et al., 2016).

**3 Methods**

**3.1 (U-Th)/He zircon thermochronology**

Low-temperature thermochronometry is a robust method to constrain the time–temperature histories of rocks (Bargnesi et al., 2016). The zircon (U-Th)/He system has a closure temperature to He diffusion of ~195-175 °C (Dodson, 1973), which provides cooling ages that can be associated with shallow processes in the crust. Additionally, the robustness of zircon to weathering

and alteration during transport and diagenesis is particularly useful in clastic systems, such as the rocks exposed in the Ellsworth Mountains. Therefore zircon (U-Th)/He dating is a powerful tool to constrain the thermal evolution of a given rock.

Zircon separates were previously prepared for the U-Pb geochronology and Hf isotope studies presented in Castillo et al. (2017), which applied standard separation procedures. Two to three single-grain aliquots from each sample were selected for (U-Th)/He analysis (Table 1). Zircon (U-Th)/He analytical methods followed those described in Guenthner et al. (2016). The details of the data reduction are shared in the Supplementary Files.

Helium extraction and analysis consisted of in vacuo diode laser heating, cryogenic purification and quadrupole mass-spectrometry on a Pfeiffer Prisma Plus at the University of Illinois. Zircon dissolution was followed by U and Th analysis via isotope-dilution inductively coupled plasma-mass spectrometry on a Thermo Element2 at the University of Arizona. Dimension measurements for zircon were collected for both the alpha ejection correction and to calculate eU concentrations. The alpha ejection correction employed the equations of Hourigan et al. (2005) and Farley (2002), with U and Th specific ejection values as listed in Farley (2002, Table 1). Depending on the degree of abrasion, one of two equations was used: tetragonal prism with pyramidal terminations (when terminations are present and measurable) or prolate spheroid (when terminations are absent). Further detail on the results of the samples and standards analysed are presented in the Table 1.

## 4 Results

### 4.1 Heritage Group

We selected three samples from the Heritage Group (Fig. 2). The samples (13EG01, 13EG05 and EHD0801A) are located in the Heritage Range in the southern sector of the Ellsworth Mountains, with an altitude that ranges from ~1450 to 990m (Table 1). Sample 13EG01 is a sandstone collected from the base of the early to middle Cambrian Union Glacier Formation and yielded ZHe ages of 179, 156 and 140 Ma (Fig. 3a). The middle to late Cambrian Springer Peak Formation (Jago and Webers, 1992; Shergold and Webers, 1992; Randall et al., 2000) is part of the upper section of the Heritage Group, from which we analysed the sandstone sample 13EG05 which yielded ZHe ages of 184, 170 and 158 Ma (Fig. 3a). The Springer Peak Formation is in lateral contact with the middle to late Cambrian Liberty Hill Formation. ZHe ages from a sandstone (sample EHD0801A) from Liberty Hill Formation yield ages of 150, 149 and 103 Ma (Fig. 3a). While most of the ZHe ages from the Heritage Group are Jurassic (from 184 to 149 Ma) two zircon grains yielded Early Cretaceous ages (140 and 103 Ma). The two younger ages are found in both the base and the upper parts of the Heritage Group, in the Union Glacier and Liberty Hills formations, respectively (Fig. 3a).

### 4.2 Whiteout Conglomerate

Two samples of matrix from conglomeratic sandstones were analysed for ZHe ages from the Whiteout Conglomerate, samples 13EG10 and 13EG15, which are located in the upper and base of this sequence, respectively. This unit was deposited during the Permian-Carboniferous (Collison et al., 1992; Matsch and Ojakangas, 1992) and is part of the upper section of the stratigraphic succession exposed in the Ellsworth Mountains. Sample 13EG10 was collected from the Whiteout Nunatak in the Sentinel Range in the northern section of the Ellsworth Mountains (Fig. 2) and yielded a ZHe age of 182 Ma (Fig. 3a). A

second rock was analysed (13EG15) from this unit to the south in the Heritage Range and to the east of the samples analysed

from the Heritage Group (Fig. 2). ZHe ages from sample 13EG15 are 791, 468 and 159 Ma (Table 1). While two grains yielded

ages older than the depositional age of this unit (~360 to 300 Ma), they agree with the provenance studies collected from the

same sample and presented by Castillo et al. (2017), who showed the presence of Palaeozoic, Neoproterozoic and

Mesoproterozoic sources in the Whiteout Conglomerate. Therefore, these results suggest that the ZHe system in the Whiteout

Conglomerate is partially reset. There is a significant distance (~250 km) between the two samples of the Whiteout

Conglomerate (Fig. 2); their altitude is 1520 and 580 m for 13EG10 and 13EG15, respectively (Table 1).

## 5 Discussion

### 5.1 Landscape evolution

Our results predominantly indicate Jurassic–Early Cretaceous ZHe ages from the Ellsworth Mountains (Fig. 3b). These results

help to constrain the landscape evolution of the Ellsworth Mountains, which is thought to have experienced an uplift event

during the Early Cretaceous (Fitzgerald and Stump, 1991) based on apatite fission-track analyses. The Jurassic–Early

Cretaceous ZHe ages are significantly younger than the age of the host sedimentary rocks, indicating the ZHe ages have been

reset. While this indicates a post-depositional thermal event, the heat source and the causative mechanism is not clear. The

Palaeozoic sedimentary sequence of the Ellsworth Mountains, although faulted and deformed (e.g. Curtis, 2001), has not

experienced significant regional metamorphism. Furthermore, sedimentary analysis shows that significant stratigraphic

repetition related to tight folding or thrust faults is not likely (e.g. Collison et al., 1992; Matsch and Ojakangas, 1992; Spörli,

1992; Webers et al., 1992a). The sedimentary sequence exposed in the Ellsworth Mountains has a thickness of ~13 km which

can account for the resetting of the ZHe ages by burial alone, assuming a geothermal gradient of 30°C km-1 and a ZHe partial

retention zone in the range of ~200-130°C (Wolfe and Stockli, 2010). Therefore, burial heating associated with deposition of

the sequence exposed in the Ellsworth Mountains may account for the resetting of the ZHe ages. It is noteworthy that the heat

flux distribution in West Antarctica is relatively complex (e.g. Martos et al., 2017). The samples are distributed along ~300

150 km (Fig. 2), which is insufficient to consider a significant change between the values of the geothermal gradient for each

sample. We nevertheless acknowledge that the chosen value may vary as there is a better understanding of the paleo-geothermal

gradient.

While all the ZHe ages in the Heritage Group range from the Jurassic to the Early Cretaceous, only two of the four grains from

the Whiteout Conglomerate are within that age range. Furthermore, the two older grains from the Whiteout Conglomerate

yield ZHe dates that are concordant with their detrital U-Pb ages, which were presented in Castillo et al. (2017), who dated the

same samples (13EG10 and 13EG15). This suggest only partial ZHe resetting occurred in the Whiteout Conglomerate and

burial heating did not reset all ZHe ages. A lesser degree of burial heating in the Whiteout Conglomerate compared to the

Heritage Group is in agreement with their respective positions in the stratigraphic sequence, in which they are towards the top

and the base of the section, respectively (Fig. 2). Zircon grains may not be fully reset in a given rock if they experience

reheating below the range of temperatures of the ZHe partial retention zone (e.g. Schneider and Issler, 2019; Malusà and Fitzgerald, 2019) as in the Whiteout Conglomerate. Conversely, zircon grains of the Heritage Group were fully reset because they experienced temperatures above the ZHe partial retention zone. Taking into the account that (i) the partial retention zone for low to moderately damaged zircon grains for the (U-Th)/He system is in the range of ~200-130°C (Wolfe and Stockli, 2010) and (ii) assuming a geothermal gradient of 30°C km-1; the Whiteout Conglomerate may have experienced between 7 – 4 km of burial. The Heritage Group experienced temperatures above ~200°C and burial by at least 7 – 6 km.

## 5.2 Gondwana fragmentation

Several tectono-magmatic events preceded the formation of oceanic lithosphere that led to the fragmentation of Gondwana and one key tectono-magmatic event is recorded in the Ellsworth-Whitmore Mountains. Nevertheless, during the Palaeozoic, the Ellsworth-Whitmore Mountains were located in the boundary between East and West Gondwana (e.g. Castillo et al., 2024; Fig. 4a). Dalziel et al. (2013 and references therein) argued that the key to understanding Gondwana's initial fragmentation in the South Atlantic-Weddell Sea region is the opposed sense of rotations of the Falklands/Malvinas Plateau and the Ellsworth-Whitmore Mountains block. These rotations were termed by Martin (2007) as 'double-saloon-door' tectonics and ascribed by them to seafloor spreading above a curved and retreating subduction zone. This event has been interpreted as an extensional episode that followed the emplacement of the Karoo and Ferrar LIPs at ~184-182 Ma (e.g. Svensen et al., 2012; Burgess et al., 2015; Greber et al., 2020) and coincides with the early development of the silicic magmatism of the Chon Aike province in the Antarctic Peninsula and Patagonia (Pankhurst et al., 2000; Bastias et al., 2021).

Grunow et al. (1987) presented a thorough revision of the paleogeographic evolution of the Ellsworth-Whitmore Mountains block based on paleomagnetic constraints. They suggested that the Ellsworth-Whitmore Mountains block and the Antarctic Peninsula have undergone little relative movement since the Middle Jurassic (Fig. 4b). Furthermore, they suggested that along with Thurston Island, the Ellsworth-Whitmore Mountains block and the Antarctic Peninsula define a single entity termed 'Weddellia'. Between the Middle Jurassic and Early Cretaceous, these crustal blocks remained attached to West Gondwana, while East Antarctica moved southward (dextrally) relative to Weddellia (Fig. 4c). The present-day position of the Ellsworth-Whitmore Mountains block was attained during the Early and mid-Cretaceous by clockwise rotation of Weddellia along with a sinistral movement relative to East Antarctica. Randall and MacNiocaill (2004) investigated the paleopositions of the Ellsworth-Whitmore Mountains block prior to the break-up of Gondwana, again based on paleomagnetic studies. Their findings suggest that the Ellsworth-Whitmore Mountains block was located near the junction of East Antarctica and Africa. However, Castillo et al. (2017), employing provenance studies, suggested a closer affinity of the Ellsworth-Whitmore Mountains block to the Australo-Antarctic plate and located this crustal block further east than that proposed by Randall and MacNiocaill (2004). Nevertheless, prior to the disassembly of Gondwana, all these studies place the Ellsworth-Whitmore Mountains block in the vicinity of the margin of the junction between East and West Gondwana, as proposed by Schopf (1969). The sequence exposed in the Ellsworth Mountains experienced burial reheating after Palaeozoic deposition. The ZHe ages presented here yield predominantly Jurassic and Early Cretaceous dates. This suggests that these rocks cooled through the

~200-130°C Zhe partial retention zone during the Jurassic and Early Cretaceous, which we interpret as exhumation related to a specific rock uplift event. Although our dataset does not contradict the seminal work of Fitzgerald and Stump (1991), who reported apatite fission-track ages ranging from ~141–117 Ma, with the exception of two zircons yielding Early Cretaceous, most of the Zhe presented herein predates 141 Ma (Fig. 3a,b). This suggests that our results provide an older age (Jurassic-Early Cretaceous) for the uplift of the Ellsworth Mountains than what was previously reported (Early Cretaceous; Fitzgerald and Stump, 1991). While this may be simply explained by the relatively higher thermal sensitivity of ZHe (~200-130°C; Wolfe and Stockli, 2010) compared to that of apatite fission-tracks (~120-60°C; Fleischer et al., 1965; Green et al., 1985), and therefore ZHe ages may potentially yield older ages than those of apatite fission-tracks, it also suggests that the Ellsworth Mountains may have uplifted earlier than previously considered, during the Jurassic. Nevertheless, this Jurassic-Early Cretaceous episode is recorded in both the Sentinel and Heritage ranges (Fig. 2), implying the presence of a regional event that affected the Ellsworth Mountains. Although the structures that uplifted these rocks are poorly constrained, the Jurassic and Early Cretaceous is dominated by widespread extension and magmatism associated with the disassembly of Gondwana (e.g. Dalziel et al., 2013; Jordan et al., 2017; Pankhurst et al., 2000; Bastias et al., 2021). We suggest that the uplift event responsible for the exhumation of the Ellsworth Mountains sequences is also part of the major plate reconfiguration associated with the break-up of Gondwana (Fig. 4b,c). An extensional setting prevailed in this sector of Gondwana during the Jurassic and Cretaceous (e.g. Dalziel et al., 2013) and therefore, we suggest that the Ellsworth Mountains were uplifted through this deformative event. However, we acknowledge that likely the Ellsworth Mountains were still being accommodated along West Antarctica during the Jurassic–Late Cretaceous (Grunow et al., 1987; Fig. 4b,c) and the uplift may have been also caused by the transtensional movement. Nevertheless, the widespread evidence of extensional tectonics in this sector of Gondwana during the Jurassic-Late Cretaceous favours an extensional mechanism.

**5.3 Connection with the Transantarctic Mountains**

**The Transantarctic Mountains extend for ~3,200 km from northern Victoria Land area of the Australian-New Zealand sector in Antarctica, to the Pensacola Mountains near the Ronne Ice Shelf (Fig. 1). They rise to elevations of >4,500 m directly from sea level along the Ross Sea coastline (e.g. Goodge, 2020). This mountain chain is a major feature of the Earth landscape, as it is the longest intraplate mountain belt. It defines the limit between East and West Antarctica, a thick stable craton and a large accretionary province, respectively (e.g. Goodge, 2020; Jordan et al., 2020). Furthermore, the Transantarctic Mountains are the world's largest rift mountain system (e.g. Goodge, 2020). Although the Ellsworth Mountains are geographically separated from the Transantarctic Mountains, both contain Precambrian and Palaeozoic rocks of similar provenance affinity (e.g. Schopf, 1969; Bradshaw, 2013) and thus they have been often correlated (e.g. Goodge, 2020). The dataset presented herein suggests that while the Transantarctic Mountains and the Ellsworth Mountains have similar rocks, their uplift may be associated with a different tectonic event. Furthermore, the thermal-tectonic history of the Transantarctic Mountains shows three major exhumation events, which occurred during the Early Cretaceous, Late Cretaceous and Cenozoic (Fitzgerald et al., 2002). These events have been associated with regional tectonic events, which are (i) the initial separation between Antarctica and Australia during the Early Cretaceous, (ii) Late Cretaceous extension (main phase) between West and East Antarctica and (iii) the Cenozoic southward seafloor propagation of the Adare Trough into the Ross Sea (e.g. Fitzgerald and Gleadow, 1988; Fitzgerald, 1992, 1994, 2002; Balestrieri et al., 1997; Miller et al., 2010; Goodge, 2020). The ZHe results presented herein**

**consistently yield Jurassic ages (nine out of eleven; Fig. 3a), which predates the initial uplift of the Transantarctic Mountains during the Early Cretaceous (e.g. Goodge, 2020). Hence, although there is some overlap in the timing of exhumation between these two mountain chains, it also suggests that the uplift of the Ellsworth Mountains may be older. Therefore, the historical correlation between the Ellsworth and Transantarctic Mountains may not be correct, at least on their uplift histories. While the uplift of the Ellsworth Mountains and the Transantarctic Mountains may be part of a diachronous exhumation event whereby uplift commenced in the Ellsworth Mountains during the Jurassic and advanced progressively towards the south, propagating into the Transantarctic Mountains during the Early Cretaceous, such a tectonic event remains unlikely considering the Gondwana breakup clockwise rotations during this period in Africa, India, Australia and New Zealand (e.g. Jokat et al., 2003). Alternatively, the Jurassic-Early Cretaceous uplift of the Ellsworth Mountains may be associated with two different tectonic episodes. First, a Jurassic uplift event occurred associated with the disassembly of southwestern Gondwana (Fig. 4b), which is exclusively recorded in the Ellsworth Mountains. A second uplift event occurred during the Early Cretaceous in both the Ellsworth Mountains and the Transantarctic Mountains (Fig. 4c). This event coincides with the first uplift of the Transantarctic Mountains, which is associated with the separation of Antarctica and Australia (e.g. Goodge, 2020).6 Conclusions**

The Palaeozoic stratigraphic sequence exposed in the Ellsworth Mountains consistently yield Jurassic and Early Cretaceous ZHe ages, which we interpret as exhumation of the Ellsworth Mountains as a direct response to rock uplift. The stratigraphic succession reached temperatures within or above the ZHe partial retention zone (~200-130°C; Wolfe and Stockli, 2010) by burial reheating associated with the ~13 km thick stratigraphic column, assuming a geothermal gradient of 30°C km-1. However, while all zircon grains from the Heritage Group yield reset ZHe ages, the Whiteout Conglomerate also yielded non-reset ZHe ages. Additionally, these non-reset ZHe ages are broadly contemporaneous with the ages of the detrital material of this unit presented by Castillo et al. (2017) on the same samples. This suggests that the temperature associated with burial reheat was, as would be expected, progressively higher towards the base of the sequence exposed in the Ellsworth Mountains. The Jurassic–Early Cretaceous uplift of the Ellsworth Mountains is older than the Early Cretaceous exhumation of the Transantarctic Mountains. We suggest that the widespread extension that dominated this sector of Gondwana and related to its fragmentation during the Jurassic and Cretaceous, is also responsible of the uplift of the Ellsworth Mountains. While the first uplift event recorded in the Transantarctic Mountains, which occurred during the Early Cretaceous, is also present in the Ellsworth Mountains, the latter also yield evidence of an older and independent Jurassic uplift episode, which we tentatively associate with the disassembly of Gondwana during this period.

**Acknowledgements**

This study was financed by project RT-4418 funded by INACH (Chilean Antarctic Survey). JB was funded by the Swiss National Science Foundation (project P5R5PN_217947). DC acknowledges support from Science Foundation Ireland (SFI) under Grant Number 13/RC/2092 and 13/RC/2092_P2 (SFI Research Centre in Applied Geosciences, iCRAG). Kei Ogata is thanked for editorial handling and the authors are grateful to three anonymous reviewers for providing constructive criticism that improved the paper.

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

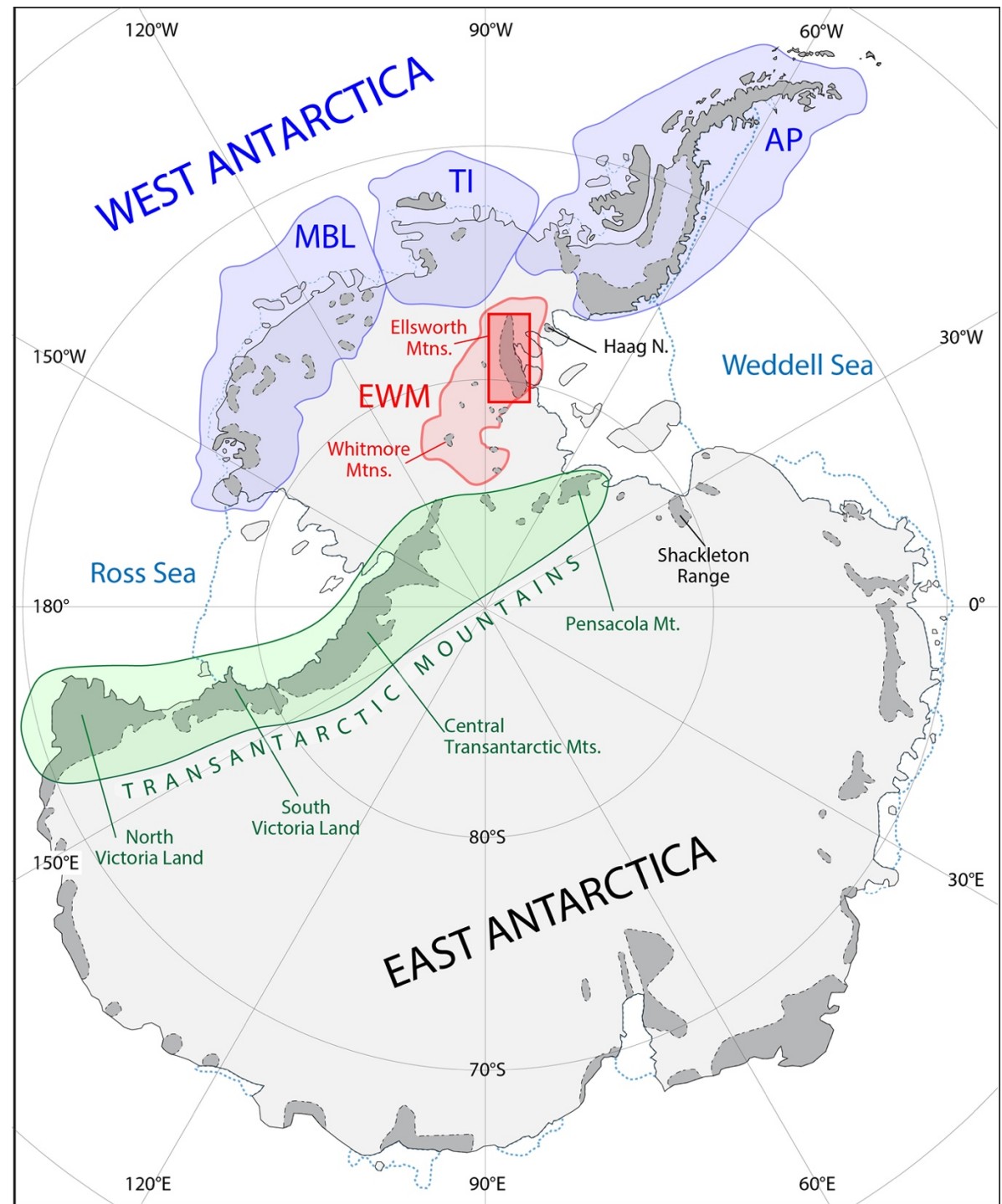

**Figure 1. Antarctica and crustal blocks of West Antarctica: AP—Antarctic Peninsula; EWM—Ellsworth-Whitmore Mountains block; MBL—Marie Byrd Land; TI—Thurston Island; N—Nunatak.**

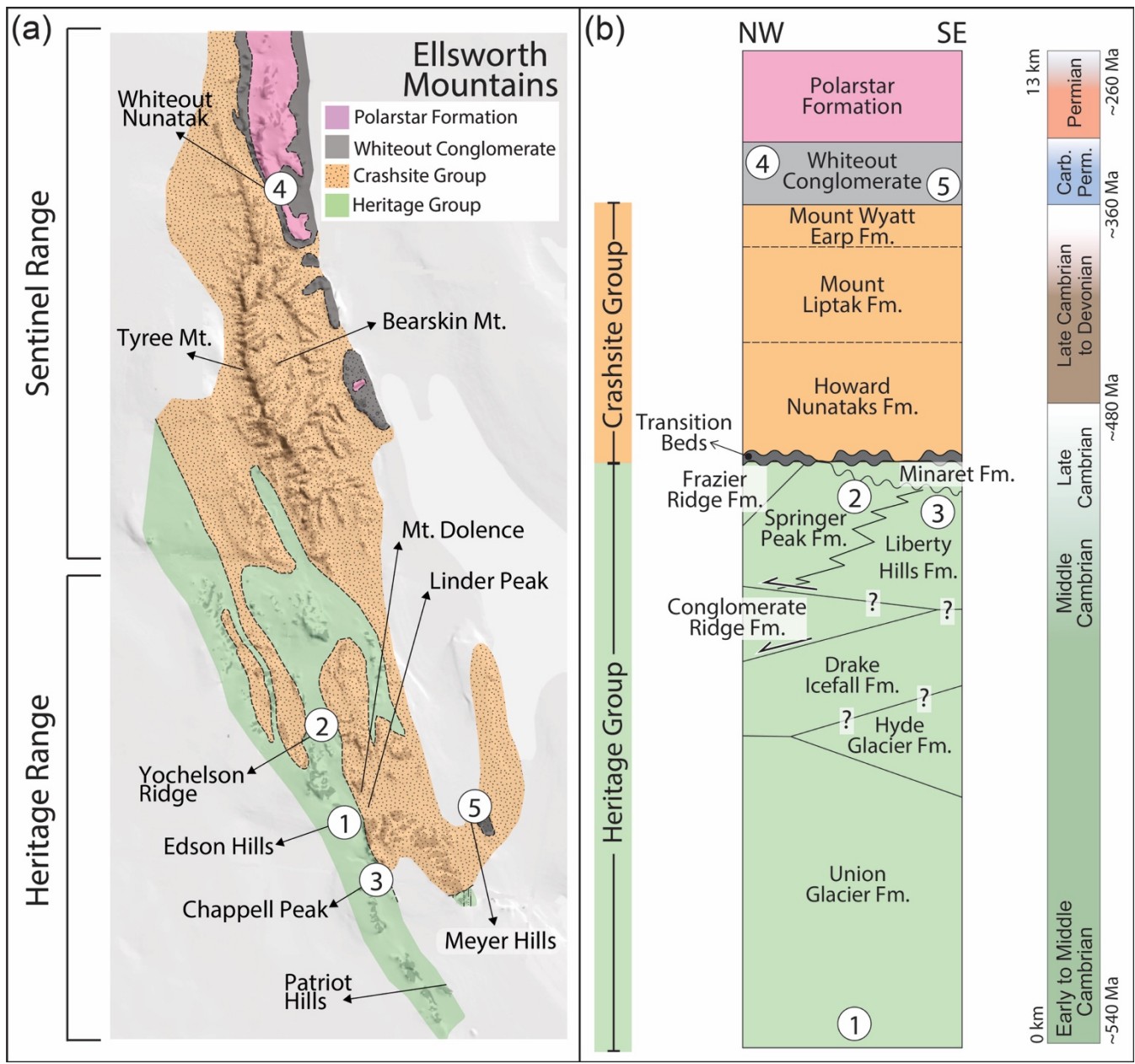

**Figure 2. (a) Simplified geological map of the Ellsworth Mountains from Craddock (1969) with the sample locations (circled numbers): 1—13EG-01; 2—13EG-05; 3—EHD0801A; 4—13EG10; 5—13EG15. (b) Stratigraphic column of the Ellsworth Mountains succession after Curtis (2001). Sample locations are place within the column according to their stratigraphic position.**

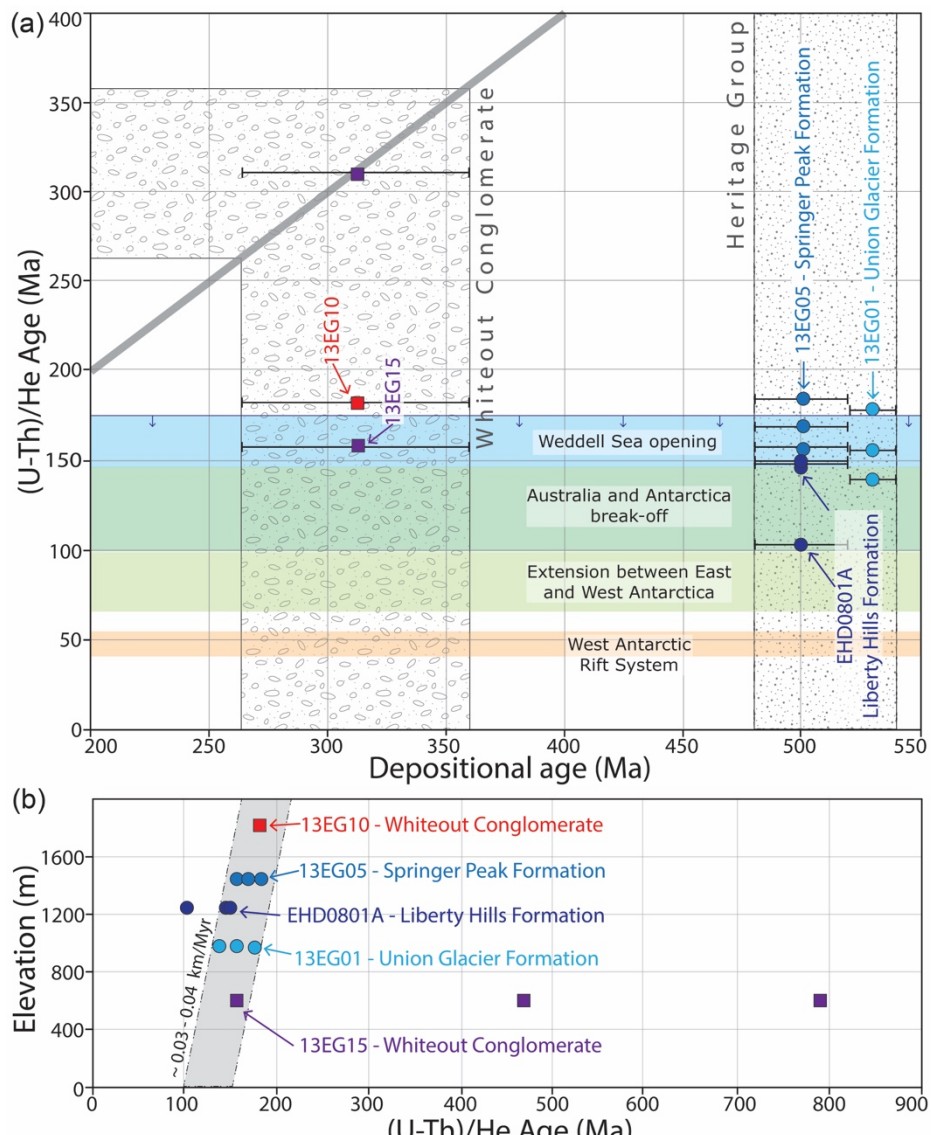

**Figure 3. (a) Depositional ages compared with ZHe ages of the grains analysed in this study. The major uplift events that formed the Transantarctic Mountains are indicated; (i) the separation between Antarctica and Australia during the Early Cretaceous, (ii) Late Cretaceous extension (main phase) between West and East Antarctica and (iii) the Cenozoic southward seafloor propagation of the Adare Trough into the Ross Sea (e.g. Fitzgerald and Gleadow, 1988; Fitzgerald, 1992, 1994, 2002; Balestrieri et al., 1997; Miller et al., 2010; Goodge, 2020) along with the timing of the Weddell Sea opening (Ghidella et al., 2002; König and Jokat, 2006). (b) Age - elevation plot of ZHe ages.**

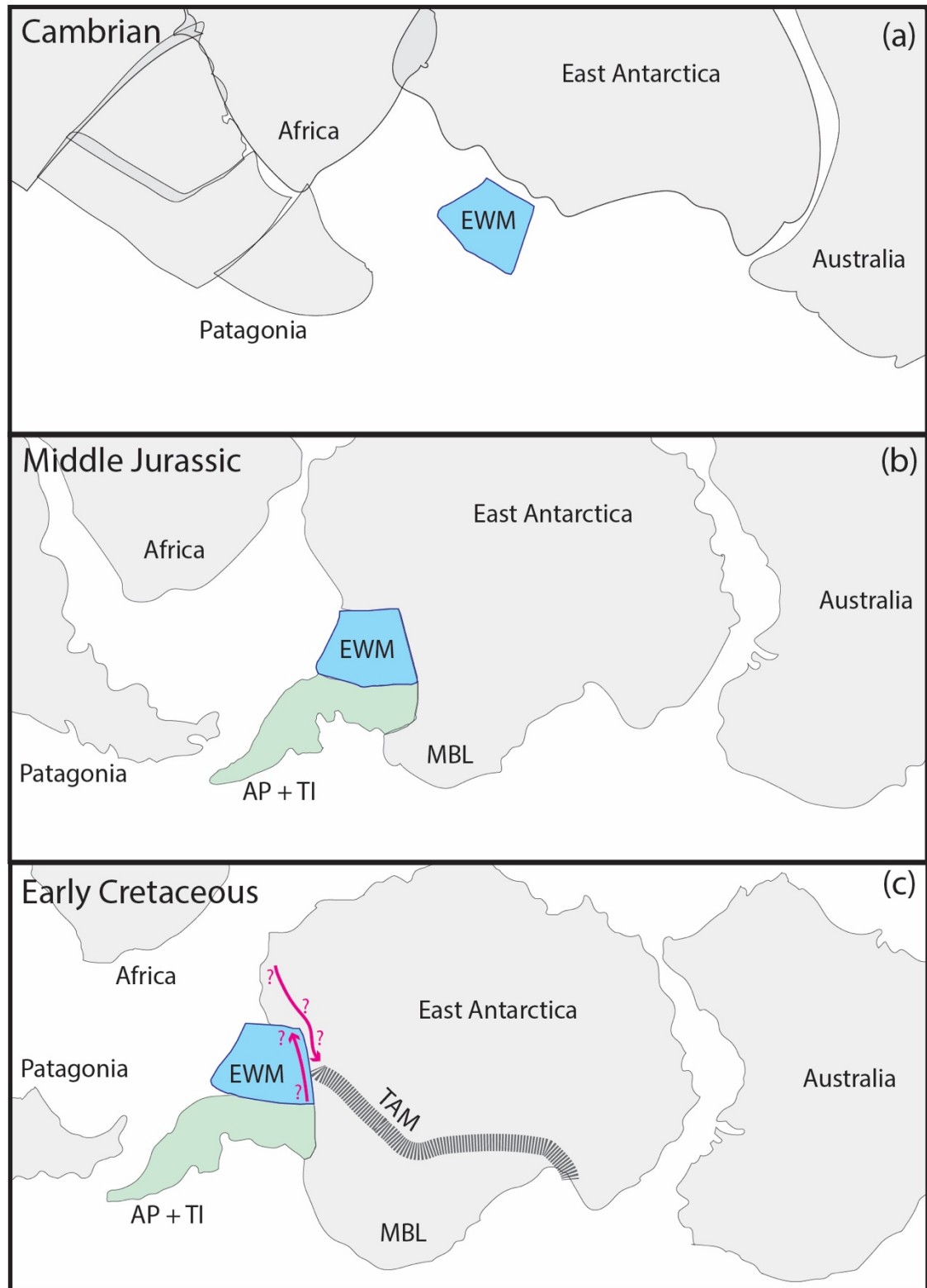

**Figure 4. Paleogeographic reconstruction of the Paleozoic and Mesozoic of southwest Gondwana, showing the position of the Ellsworth-Whitmore Mountains relative to the surrounding crustal blocks. The boundaries of the crustal blocks are schematic. (a) Cambrian (extracted from Castillo et al., 2024), the Ellsworth-Whitmore Mountains are located between East and West Gondwana without experiencing relevant deformation. (b) Middle Jurassic (extracted from Grunow et al., 1987), the Ellsworth-Whitmore Mountains experience little relative motion with respect to East Antarctica. We associate this with the initiation of the uplift of the Ellsworth-Whitmore Mountains. (c) Early Cretaceous (extracted from Grunow et al., 1987), the Ellsworth-Whitmore Mountains experienced a notable relative movement with respect to East Antarctica, which consisted of ~650 km of dextral strike-slip displacement. We suggest that throughout this significant displacement, deformation continued and intensified in the Ellsworth-Whitmore Mountains, which finished to uplift this mountain chain. The dotted line corresponds to the Transantarctic Mountains. AP + TI: Antarctic Peninsula and Thurston Island. EWM: Ellsworth-Whitmore Mountains. MBL: Marie Byrd Land. TAM: Transantarctic Mountains.**

**Table 1. Single grain zircon (U-Th)/He dating results from the Heritage Group and Whiteout Conglomerate.**

| Sample and aliquots | South | West | Altitude | Lithology | Group | Unit | Stratigraphical Age* | 4He (pmol) | ± | U (ng) | ± | Th (ng) | ± | Rs[b] (µm) | Mass (g) | 4He (nmol/g) | ± | U (ppm) | ± | Th (ppm) | ± | eU (ppm) | Uncorr date (Ma) | ±2σ (Ma) | F_T | Corrected date (Ma) |
|---|---|---|---|---|---|---|---|---|---|---|---|---|---|---|---|---|---|---|---|---|---|---|---|---|---|---|
| **13EG01** | -79.80 | -83.65 | 987 | Sandstone | Heritage Group | Union Glacier Formation | early Cambrian (~540-520 Ma) | | | | | | | | | | | | | | | | | | | |
| z01 | | | | | | | | 0.54 | 0.0033 | 0.88 | 0.013 | 0.30 | 0.004 | 49 | 0.00271 | 129 | 0.8 | 210 | 3.0 | 72 | 1.0 | 227 | 104.6 | 3.0 | 0.8 | 139.3 |
| z02 | | | | | | | | 0.33 | 0.0021 | 0.39 | 0.006 | 0.34 | 0.005 | 45 | 0.00213 | 101 | 0.6 | 118 | 1.7 | 103 | 1.5 | 142 | 130.5 | 3.5 | 0.7 | 178.2 |
| z03 | | | | | | | | 0.58 | 0.0035 | 0.76 | 0.011 | 0.45 | 0.006 | 58 | 0.00392 | 86 | 0.5 | 113 | 1.6 | 67 | 1.0 | 129 | 123.0 | 3.4 | 0.8 | 155.8 |
| **13EG05** | -79.61 | -84.45 | 1443 | Sandstone | Heritage Group | Springer Peak | middle- to late Cambrian (~520-480 Ma) | | | | | | | | | | | | | | | | | | | |
| z01 | | | | | | | | 0.62 | 0.0038 | 0.72 | 0.010 | 0.42 | 0.006 | 49 | 0.00309 | 133 | 0.8 | 155 | 2.2 | 91 | 1.3 | 177 | 138.2 | 3.8 | 0.8 | 184.0 |
| z02 | | | | | | | | 0.21 | 0.0009 | 0.29 | 0.004 | 0.15 | 0.002 | 41 | 0.00165 | 84 | 0.3 | 115 | 1.6 | 58 | 0.8 | 128 | 119.9 | 3.2 | 0.7 | 169.2 |
| z03 | | | | | | | | 0.56 | 0.0023 | 0.78 | 0.011 | 0.42 | 0.006 | 48 | 0.00230 | 144 | 0.6 | 200 | 2.8 | 107 | 1.5 | 225 | 117.6 | 3.1 | 0.7 | 157.5 |
| **EHD0801A** | -79.97 | -82.94 | 1240 | | Heritage Group | Liberty Hills Formation | middle- to late Cambrian (~520-480 Ma) | | | | | | | | | | | | | | | | | | | |
| z01 | | | | | | | | 0.14 | 0.0006 | 0.22 | 0.003 | 0.12 | 0.002 | 41 | 0.00137 | 64 | 0.3 | 99 | 1.4 | 57 | 0.8 | 112 | 105.5 | 2.8 | 0.7 | 149.7 |
| z02 | | | | | | | | 0.02 | 0.0001 | 0.05 | 0.001 | 0.01 | 0.000 | 40 | 0.00004 | 8 | 0.0 | 20 | 0.3 | 5 | 0.1 | 21 | 72.2 | 2.1 | 0.7 | 102.9 |
| z03 | | | | | | | | 0.11 | 0.0005 | 0.18 | 0.003 | 0.08 | 0.001 | 37 | 0.00090 | 71 | 0.3 | 118 | 1.7 | 54 | 0.8 | 130 | 100.6 | 2.7 | 0.7 | 148.3 |
| **13EG10** | -77.60 | -86.32 | 1810 | Conglomeratic sandstone (matrix) | | Whiteout Conglomerate | Permain - Carbonifeours (~360-260 Ma) | | | | | | | | | | | | | | | | | | | |
| z03 | | | | | | | | 1.13 | 0.0047 | 1.61 | 0.023 | 0.14 | 0.002 | 39 | 0.00201 | 466 | 1.9 | 662 | 9.4 | 57 | 0.8 | 676 | 126.6 | 3.6 | 0.7 | 181.7 |
| **13EG15** | -79.77 | -81.30 | 600 | Conglomeratic sandstone (matrix) | | Whiteout Conglomerate | Permain - Carbonifeours (~360-260 Ma) | | | | | | | | | | | | | | | | | | | |
| z01 | | | | | | | | 2.65 | 0.0109 | 3.80 | 0.054 | 0.39 | 0.006 | 58 | 0.00508 | 324 | 1.3 | 465 | 6.6 | 47 | 0.7 | 476 | 124.9 | 3.5 | 0.8 | 158.2 |

| | | | | | | | | | | | | | | | | | | | | | | | | | | |
|---|---|---|---|---|---|---|---|---|---|---|---|---|---|---|---|---|---|---|---|---|---|---|---|---|---|---|
| z02 | | | | | | | | 0.77 | 0.0035 | 0.37 | 0.005 | 0.05 | 0.001 | 50 | 0.00029 | 156 | 0.7 | 76 | 1.1 | 11 | 0.2 | 78 | 357.4 | 10.5 | 0.8 | 467.2 |
| z03 | | | | | | | | 1.16 | 0.0053 | 0.33 | 0.005 | 0.00 | 0.000 | 51 | 0.00001 | 222 | 1.0 | 63 | 0.9 | 1 | 0.0 | 63 | 614.2 | 18.8 | 0.8 | 790.4 |
| | | | | | | | | | | | | | | | | | | | | | | | | | | |
| **Standard** | | | | | | | | | | | | | | | | | | | | | | | | | | |
| FCTjm_Zr14 | | | | | | | | 0.14 | 0.0007 | 1.04 | 0.015 | 0.88 | 0.013 | 41 | 0.00158 | 63 | 0.3 | 455 | 6.6 | 387 | 5.6 | 546 | 21.4 | 0.5 | 0.7 | 30.3 |
| FCTjm_Zr13 | | | | | | | | 0.19 | 0.0012 | 1.42 | 0.020 | 0.82 | 0.012 | 43 | 0.00224 | 67 | 0.4 | 509 | 7.3 | 295 | 4.2 | 578 | 21.5 | 0.6 | 0.7 | 30.0 |

* Stratigraphical ages have been estimated from the column presented in Curtis (2001)

## Code/Data availability

All raw unprocessed data related to this study can be requested by contacting the corresponding author.

## Author Contribution

Joaquín Bastías-Silva: conceptualisation, leading manuscript development, leading data interpretation, visualisation.

David Chew: manuscript development, visualisation.

Fernando Poblete: funding acquisition, conceptualisation, sample collection, manuscript development.

Paula Castillo: conceptualisation, sample collection, manuscript development.

William Guenthner: data collection, manuscript development.

Anne Grunow: manuscript development.

Ian W.D. Dalziel: manuscript development.

Airton N. C. Dias: manuscript development.

Cristóbal Ramírez de Arellano: sample collection.

Rodrigo Fernandez: conceptualisation, sample collection.

## Competing interest

The (co-)authors are not member of the editorial board of Solid Earth and/or a guest member of the editorial board of Solid Earth. The peer-review process will be guided by an independent editor, and the authors also have no other competing interests to declare.