# Peer review of "Uplift and denudation history of the Ellsworth Mountains: insights from low temperature thermochronology"

_EGUsphere, 2023_

## Author Response (AR1)

Review of "Uplift and denudation history of the Ellsworth Mountains: insights from low temperature thermochronology". This paper presents new evidence for when the elevated topography of the Ellsworth Mountains was first uplifted. The recovered cooling dates are generally late Jurassic to Early Cretaceous in age. Isolated older dates associated with stratigraphically shallower rocks are consistent with the temperature thermochronology not being fully reset by burial. The authors conclude that the main phase of uplift occurred during the Jurassic breakup of Gondwana which initiated in the Weddell Sea region.

Overall the paper seems well written and the conclusions are supported by the data given. I have a couple of suggestions for the authors to consider.

I hope the authors find these suggestions useful.

First: L184-186 the authors note that "This suggests that these rocks cooled through the ~200-130°C ZHe retention zone during the Jurassic and Early Cretaceous, which we interpret as exhumation related to a specific rock uplift event which supports the seminal work of Fitzgerald and Stump (1991)". It is correct that the results do not contradict the work of Fitzgerald and Stump (1991), as no upper limit (prior to 141 Ma) was given in the 1991 paper. However, almost all the new results pre-date the 141 Ma given in the previous paper, and only two are Cretaceous, while 9 are Jurassic (excluding older outliers). I think the authors could reasonably argue that their results provide an older, but not inconsistent, age for the uplift of the Ellsworth Mountains compared to Fitzgerald and Stump (1991). I think this is worth saying, as the top line 140 Ma age provided in the 1991 paper suggests a more 'Transantarctic'/Cretaceous origin for the uplift, while the provided data seems to point more to an older Jurassic/Weddell Sea breakup origin.

Answer: The text has been modified accordingly.

Second: The authors suggest the Ellsworth Mountains contain a distinct Jurassic uplift signal compared with the rest of the Transantarctic Mountains, and argue quite convincingly that this is associated with Weddell Sea rifting. However, on L217 they suggest and alternative model where by "the uplift of the Ellsworth Mountains and the Transantarctic Mountains may be part of a diachronous exhumation event whereby uplift commenced in the Ellsworth Mountains during the Jurassic and advanced progressively towards the south, propagating into the Transantarctic Mountains during the Early Cretaceous". It is well understood that Gondwana breakup progressed clockwise splitting Africa, India, Australia and New Zealand away from Antarctica. A diachronous event moving in the opposite direction, although not possible to totally rule out, seems less likely, and maybe this sentence should be caveated. More likely two uplift events impacted the Ellsworth Mountains, one in the Jurassic associated with the Weddell Sea, and a secondary overprint in the Cretaceous associated with the Wider Transantarctic Mountain system.

Answer: The text has been modified accordingly.

Review of "Uplift and denudation history of the Ellsworth Mountains: insights from low-temperature thermochronology" by Bastias-Silvas and Coauthors.

Bastias-Silvas and coauthors present new zircon (U-Th)/He (ZHe) age data from rock samples near the base and top of the Ellsworth Mountains, a part of West Antarctica. Detrital ZHe ages from the Neoproterozoic – Permian sedimentary pile reveal that the rocks experienced burial after deposition with a characteristic cooling event in the Jurassic to Early Cretaceous.  Based on ZHe system thermochronological constraints, the authors conclude that the uplift and exhumation of the Ellsworth Mountains during the Jurassic to Early Cretaceous coincided with the rotation and translation of this crustal block relative to East Antarctica. This period corresponds with widespread extension associated with the breakup of Gondwana, with the Ellsworth Mountains playing a significant role in the opening of the far South Atlantic. Furthermore, the study indicates that the uplift of the Ellsworth Mountains started in the Jurassic period, predating the Early Cretaceous uplift of the Transantarctic Mountains. This observation suggests that continental-scale, rift-related exhumation was not synchronous, with the initiation in the Ellsworth Mountains in the Jurassic and later propagating southwards into the Transantarctic Mountains during the Early Cretaceous.

While the paper provides valuable insights into the long-term landscape evolution of the Antarctic region, Gondwana fragmentation, and the connection with the Transantarctic Mountains in the Ellsworth Mountains, there are a few points which need attention.

1)    Extension is a rather general mechanism which indeed could provide uplift of the footwall rocks concerning a main or a set of normal faults. The paper text and the figures fail to highlight a possible set of structures responsible for such a mechanism. I believe that the paper would benefit from an additional figure where the author's effort to delineate the tectonic setting driving the major phase of exhumation.

Answer: The reviewer has raised a fair point. We have also debated this and yet we are not able to provide a consistent solution for the reader, which is the reason why we don't further address this in the manuscript. While we agree that this period is vastly dominated by extensional tectonics, the particular mechanism responsible for the uplift of the Ellsworth Mountains remains to be proven. Therefore, we think that it may be either (i) an extensional or (ii) a local compressive structure, the latter is probably a result of the overall accommodation of the crustal blocks from this sector in Gondwana during this period. We consider that providing a mechanism, which we know may be easily disproved, is not beneficial for the manuscript or the reader. We are nevertheless working on a follow-up study, which will address this point.

2)   Assumption of the geothermal gradient: The authors based the main Early Cretaceous-Jurassic cooling event of the ~13 km thick sedimentary pile on the assumption of a 30°C geothermal gradient. However, West Antartica has a very complex pattern of heat flux (e.g., Martos et al., 2017), how reliable is the assumed thermal gradient and how spatial variability of the heat flux would caveat major assumption (i.e., sufficient time for partial annealing of the ZHe system)?

Answer: The text has been modified to better explain these questions along with indicating the complex nature of the heat flow of Antarctica.

3)   The discussion part needs a diagram or figure, later in the text I suggest a map view figure and/or model describing the tectonic evolution of the area in a graphic form.

Answer: We disagree. We are not able to provide such a material at the moment. This is not directly related to the study but rather to the nature of the location, which is very remote and therefore the available information is scarce. In this scenario, such a figure, would be speculative.

I provide a few additional points of discussion that I would recommend considering before publication.

Intro:

Please check the punctuation throughout the introduction, the use of the commas is not always correct: e.g., line 36: ... Antarctic Peninsula, and are ~50...

Answer: The text has been modified accordingly.

Methods:

Overall, the method section could benefit from additional information, such as the specific parameters used during the helium extraction and the isotope-dilution process. Nor in the current or cited Castillo et al., 2017 is possible to know the grain size of the analyzed zircons. Yet, there is no mention of the statistical treatment of data, which is crucial for the interpretation of (U-Th)/He ages and for the discussion of the data in the geodynamic evolution of the area. Additionally, it would be useful to include a brief description of how dimension measurements for zircon were collected and how the alpha ejection correction was applied, as these details are critical for understanding the analytical process. In Figure 3, the standard error of the age is not plotted. Please add the information or refer to the corrected age that you seem to plot. The authors often refer to Table 1, however, the text should reference this table to guide readers to the relevant data.

Answer: We do not understand very well what the reviewer is pointing out. The method section does reference Guenthner et al. (2016), which contains all the details of the methods in it. We are not sure what the reviewer means with 'specific parameters'. The information in terms of instrumentation is relatively specific and that text is pretty standard for a (U-Th)/He methods section. Uncertainties were propagated using a script for the data reduction developed in the lab, which is present on the lab's GitHub website here:

HAL_data/He_date_calc.ipynb at main · wrguenthner/HAL_datagithub.com

A copy of this script will be shared on the Supplementary Files of the article.

For the gran size measurements, we report the spherical radius (Rs), as it has been standardised in the literature. Alpha ejection correction was performed using the equations described in Reiners et al. (2005) and Hourigan et al. (2005), with the U and Th specific ejection values as listed in Farley (2002), Table 1. Two equations were used, depending upon the degree of abrasion: tetragonal prism with pyramidal terminations (terminations are present and measurable), or prolate spheroid (terminations are absent).

Standard error is not observed in Figure 3 because the plotted points (icons) overlap standard error. Therefore, if plotted, it would be a graphic overestimation of the standard errors. They are nevertheless shared on Table 1.

We have modified the text and complemented it with these observations.

In line 95, there is a redundancy with the incipit of the paragraph. ZHe is indeed a robust technique,; it would be more honest to state that multi-proxy thermochronology (i.e., comparing thermochronometers with different closure ranges) is now accepted as a standard procedure that can help to better understand the time–temperature evolution of a given landscape.

Answer: The text has been modified accordingly.

Results section:

Overall the result section is well structured and can be followed by non-experts of the area. Yet there is a point that could be improved and there is the lack of detailed explanation on why elevation profiles have a significance or not in the present set of data, there is no mention of this. The samples are hundreds of km apart one from the other, does the spatial distance among samples may or not have an impact on the correct data interpretation?

Answer: While the altitude is not relevant for the results of this study, the stratigraphic position of the samples is critical.

On the second observation, it is mentioned that the results are consistent throughout all the dataset, which we use to suggest that the Jurassic-Early Cretaceous uplift is a regional event (line 200 of the corrected version).

We have complemented the text with these observations.

Discussion:

The assumption on geothermal gradient in West Antarctica: Is it reasonable to assume an average of ~30°C km-1 as you mention in lines 144-145? The authors should specify on how the variability of geothermal gradient and heat flux can affect their calculation as it can vary in the order of10 to 20 0° C km-1 (e.g., Martos et al., 2017).

Answer: The text has been modified to better explain these questions along with indicating the complex nature of the heat flow of Antarctica.

Sections 5.2 and 5.3 would need a figure where the major geodynamic phases are described in map view. These parts of the text lack an accurate discussion of potential causal mechanisms for the recorded geological events. It's essential to delve deeper into the potential geological processes and their implications for the region's tectonic histor. Thiss can be followed much better by looking at a summary figure where the major events are described as a graphic. I strongly believe that the quality of the paper will improve with such a diagram.

Answer: We disagree. We are not able to provide such a material at the moment. This is not directly related to the study but rather to the nature of the location, which is very remote and therefore the available information is scarce. In this scenario, such a figure, would be speculative.

Line-by-line comments:

185: partial retention zone.

Answer: The text has been modified accordingly.

195: how would a transtensional movement produce a significant uplift able to reset a pile of sediments from burial T of ~200°C to the surface? Please explain.

Answer: We would prefer to not further elaborate on this point, as we do not have sufficient field data to robustly provide a clear mechanism. While we could elaborate examples from the literature in other locations, at this point, we do not have the necessary information to choose a particular mechanism to be presented

for the reader. Additionally, this may change as soon as there is further field data. Therefore, we have intentionally omitted this point.

2015-2020: Along what tectonic structures such southward propagation would work? Please explain.

Answer: We have changed this interpretation accordingly with your observations and the other reviewer.

Figure 1: The authors should indicate the trace of the major fault systems bounding the Ellsworth and Transatlantic Mountains.

Answer: We have shared the best geological map available. Given the location and difficult access, a structural map lacks on the region.

References:

Martos, Y. M., Catalán, M., Jordan, T. A., Golynsky, A., Golynsky, D., Eagles, G., &Vaughan, D. G. (2017). Heat flux distribution of Antarctica unveiled.Geophysical Research Letters,44,11,417–11,426. https://doi.org/10.1002/2017

---

## Author Response (AR2)

Referee # 2

Thank you for the revision. As stated in the previous comments, I maintain my position that the discussion would benefit a figure wherein the authors aim to summarize the main interpretations derived from their dataset (i.e., Landscape evolution). I am not fully convinced by your response, as I believe that any graphic is more accessible than delving into a paper text to follow your interpretations. If I were in your position, I would have attempted to frame the time of interest with a tectonic reconstruction to provide a hint on where and when major structures could be responsible for the two exhumation phases.

However, I get the impression that it may not be in your interest to speculate and consolidate these tectonic observations into a model that can provide context for this differential exhumation to a larger audience. I do not have any further feedback.

Best Wishes.

Answer:

Dear reviewer,

Although we initially disagree, we have followed your advice and added a figure (Figure 4) summarising our interpretation of the tectonics events that are responsible for the uplifting of the Ellsworth-Mountains.

Best wishes,
The authors

Referee #3

This manuscript presents new thermochronological data from the Ellsworth Mountains. This is an exciting region of general importance. However, as it stands now, the study requires major revisions.

The manuscript is a short communication on new ages. Based on the observed ages and the observation that the ages are partially reset in the upper part of the stratigraphic column and completely reset at the base, the authors argue that a major rearrangement of plates occurred earlier than previously known.

The presented data are of high quality. However, as it stands now the conclusions remain rather speculative. Standard age-elevation plots are missing, which would make it much easier to evaluate the results.

Answer: We have followed your advice and have added an age-elevation plot (Fig. 3b).

Different estimates of exhumation are presented but only as rough calculations. Instead, numerical t-T models should be performed. This could lead to very informative results, especially since parts of the ages are not completely reset. This should be accompanied by a more extensive discussion of the spread in single grain ages, particularly for the completely reset grains.

Answer: Although we agree that this is an interesting point, and it would be beneficial for the article, it is rather difficult to accomplish such a suggestion. This is mainly given by the fact that our study only has (U-Th)/He data, and t-T models of basins require further thermochronological constraints (e.g. Cogné et al., 2012; Jess et al., 2018; Prenzel et al., 2018; Gallagher & Parra, 2020; Krob et al., 2020; Licciardi et al., 2020; Wildman et al., 2021). Additionally, unfortunately, we cannot complement this with the literature, as there is no such information available. Therefore, we prefer to avoid modelling the t-T paths of the dataset, as it is.

- *Cogné, N., Gallagher, K., Cobbold, P. R., Riccomini, C., & Gautheron, C. (2012). Post-breakup tectonics in southeast Brazil from thermochronological data and combined inverse-forward thermal history modeling. Journal of Geophysical Research: Solid Earth, 117(B11).*
- *Gallagher, K., & Parra, M. (2020). A new approach to thermal history modelling with detrital low temperature thermochronological data. Earth and Planetary Science Letters, 529, 115872.*
- *Jess, S., Stephenson, R., & Brown, R. (2018). Evolution of the central West Greenland margin and the Nuussuaq Basin: Localised basin uplift along a stable continental margin proposed from thermochronological data. Basin Research, 30(6), 1230-1246.*

- *Krob, F. C., Glasmacher, U. A., Bunge, H. P., Friedrich, A. M., & Hackspacher, P. C. (2020). Application of stratigraphic frameworks and thermochronological data on the Mesozoic SW Gondwana intraplate environment to retrieve the Paraná-Etendeka plume movement. Gondwana Research, 84, 81-110.*
- *Licciardi, A., Gallagher, K., & Clark, S. A. (2020). A Bayesian approach for thermal history reconstruction in basin modeling. Journal of Geophysical Research: Solid Earth, 125(7), e2020JB019384.*
- *Prenzel, J., Lisker, F., Monsees, N., Balestrieri, M. L., Läufer, A., & Spiegel, C. (2018). Development and inversion of the Mesozoic Victoria Basin in the Terra Nova Bay (Transantarctic Mountains) derived from thermochronological data. Gondwana Research, 53, 110-128.*
- *Wildman, M., Gallagher, K., Chew, D., & Carter, A. (2021). From sink to source: Using offshore thermochronometric data to extract onshore erosion signals in Namibia. Basin Research, 33(2), 1580-1602.*

The tectonic history as well as the connection to the Transantarctic Mountains remain hypothetical. Suggestions from an earlier review to present more details on regional geology and discuss potential driving mechanisms have not been accommodated. These seem however crucial to support the conclusions. Similarly, the request for sketches of how the tectonic scenario is envisioned has not been accommodated. Such (maybe even only conceptual) sketches would greatly help understand the presented story.

Answer: We have followed your advice and added a figure that summarise the tectonic evolution of the Ellsworth Mountains (Figure 4). We have also modified the text accordingly.

In summary, this study is well written and has great potential with a sound data set from an interesting region. However, major revisions are required before this study can be published.

Answer: We have tried to modify the text and the figures accordingly to your advice. We appreciate the review.